Serum brain-derived neurotrophic factor, micronutrient status, and inflammatory cytokines in type 2 diabetes with nephropathy: a case–control analysis

Waheed Huda Jaber dr.huda.jw@uomustansiriyah.edu.iq 1
Numan Nawfal A. 2
1 Department of Pharmacology and Toxicology, Mustansiriyah University , Baghdad , Iraq
2 Department of Pharmacology and Toxicology, Mashreq University , Baghdad , Iraq
Uversky Vladimir
Electronic publication date: 2025 Oct 20
Publication date: 2025
Volume: 13
Electronic Location ID: e20086
Received 2025 Apr 28; Accepted 2025 Aug 26
Copyright: ©2025 Waheed and Numan
Copyright year: 2025
Copyright holder: Waheed and Numan
License: This is an open access article distributed under the terms of the Creative Commons Attribution License, which permits unrestricted use, distribution, reproduction and adaptation in any medium and for any purpose provided that it is properly attributed. For attribution, the original author(s), title, publication source (PeerJ) and either DOI or URL of the article must be cited.
License URL: https://creativecommons.org/licenses/by/4.0/

Keywords: Diabetes mellitus, Diabetic nephropathies, Brain-derived neurotrophic factor, Diabetes complications

Funding: The authors received no funding for this work.

==============================
Background

Brain-derived neurotrophic factor (BDNF) levels are lower in diabetic patients compared to healthy individuals, and may be further affected by nephropathy. This study aimed to evaluate serum BDNF levels in diabetic patients with nephropathy without complications and compare them to levels in healthy control subjects.

Methods

A case–control study was conducted involving three groups: healthy individuals (controls), diabetic patients without complications (DM), and diabetic nephropathy patients (DN). Serum BDNF levels were measured using the sandwich Enzyme-Linked Immunosorbent Assay (ELISA) technique, alongside serum levels of interleukin (IL)-12, IL-16, folic acid, vitamin B12, vitamin D3, total cholesterol, triglycerides, high density lipoprotein (HDL), hemoglobin A1c (HbA1c), urea, creatinine, total calcium, and zinc.

Results

Serum BDNF levels were significantly lower in the DN group compared to the DM and control groups (42.1, 34.1, and 27.23 ng/mL, respectively). In the DM group, BDNF showed a direct correlation with HbA1c and urea (r = 0.26 and r = 0.35, respectively), and an inverse correlation with fasting plasma glucose (FPG) (r =  − 0.30). In the DN group, BDNF was directly correlated with FPG (r = 0.31) and serum creatinine (r = 0.27). The area under the curve (AUC) for BDNF in distinguishing DN from controls was 0.938, and 0.738 for DN versus DM.

Conclusion

Serum BDNF levels are markedly reduced in type 2 diabetic patients with nephropathy and correlate with deficiencies in vitamins D, B12, folate, and zinc, as well as elevated IL-6 and IL-12. BDNF may serve as a biomarker for diabetic kidney disease, highlighting the importance of nutritional status, inflammation control, and neurotrophic support.

Introduction

Diabetes mellitus is a chronic metabolic disorder marked by hyperglycemia and systemic complications. Over time, diabetes can impair multiple organ systems, with diabetic nephropathy being a common microvascular complication (Antar et al., 2023; Numan, Jawad & Fawzi, 2024b). Nutritional factors, such as vitamins and minerals, as well as inflammatory mediators, play pivotal roles in the progression of diabetes and its complications. Key micronutrients, including vitamin D, vitamin B12, folic acid, zinc, and calcium, are often disrupted in diabetic patients and can impact glucose metabolism, immunity, and vascular health (Wen et al., 2006; Younes, 2024; Numan, Jawad & Fawzi, 2024a; Atia et al., 2024). Concurrently, pro-inflammatory cytokines such as interleukin-6 (IL-6) and interleukin-12 (IL-12) are frequently elevated in diabetes, contributing to insulin resistance and tissue damage (Kadhim et al., 2018). Another emerging element is brain-derived neurotrophic factor (BDNF), a neurotrophin involved in both neural and metabolic regulation, which appears to be altered in diabetic conditions (Senn et al., 2002; Nashtahosseini et al., 2025).

Diabetic nephropathy (DN) is a common complication of diabetes. It is a slow and progressive kidney disease marked by kidney damage, leading to the failure of the glomerular filtration capacity (Rout & Jialal, 2025). DN leads to kidney impairment in a vast majority of diabetic patients, and long-term hyperglycemia, the hallmark of diabetes, further prompts the development of other complications (Sagoo & Gnudi, 2020). As a result of such a high prevalence of diabetes worldwide, patients who suffer from chronic kidney disease will experience a drastic increase (Hoogeveen, 2022; Ghassan, Jaber & Hassan, 2020).

Brain-derived neurotrophic factor (BDNF), which is most abundant in the central nervous system (Albini, Krawczun-Rygmaczewska & Cesca, 2023). BDNF is essential for the survival, growth, and maintenance of the neurodevelopment of the nervous system and the cognitive signaling required for related tissues and cells (Bathina & Das, 2015). Furthermore, BDNF crosses the blood–brain barrier as an endocrine hormone and circulates throughout the body (Banks, 2012). Plasma BDNF is crucial in maintaining healthy cognitive function (Merighi, 2024). Because it is easy to obtain by non-invasive blood sampling, it can be measured to assess cognitive function or used as a disease-monitoring indicator (Komulainen et al., 2008). Diabetes is associated with changes in levels of BDNF, which leads to a plethora of neurological complications, highlighting the importance of transitioning through the health perspectives of a DN patient (Sumbul-Sekerci et al., 2023). Several studies have shown that BDNF may be associated with other kinds of diabetic complications. The brains of diabetic patients face a heightened risk of microangiopathy resulting from the coexistence of complications associated with both type 1 and type 2 diabetes. Cerebral microangiopathy contributes to cognitive decline, gait disturbances, and increased stroke risk (Gaur et al., 2024; Mauricio, Gratacòs & Franch-Nadal, 2023). Impairment in endothelial functioning hinders microcirculation and nutritive blood flow to neural tissue, particularly in the retina and the brain (Gaur et al., 2024).

BDNF and its principal receptor are expressed in the pancreas, and both have been the focus of extensive investigation in this organ. Research has shown that BDNF signaling supports the survival of pancreatic β-cells and insulin secretion. Studies investigate how BDNF–TrkB interactions impact glucose homeostasis and metabolic regulation. Altered BDNF levels in diabetic models suggest therapeutic potential for preserving islet function (Zhu et al., 2001; Fulgenzi et al., 2020). Circulating levels of BDNF have been measured in plasma, serum, and serum exosomes of diabetic and healthy subjects (Moosaie et al., 2023).

Plasma and serum BDNF levels are downregulated in patients with type 1 and type 2 diabetes (Rozanska, Uruska & Zozulinska-Ziolkiewicz, 2020; He et al., 2024). Still, the number of studies has not been as extensive as those evaluating retinopathy and nephropathy. In type 1 diabetes, disease duration has been inversely associated with plasma BDNF levels (Uruska et al., 2017). Plasma BDNF decreases with indicators of renal function, such as decreased glomerular filtration rate and increased urinary albumin excretion, as well as duration and severity of illness in type 2 diabetes (Rozanska, Uruska & Zozulinska-Ziolkiewicz, 2020).

Vitamin D is not only essential for calcium homeostasis but also modulates insulin secretion and immune responses (Alqudah et al., 2022). Diabetes is frequently associated with vitamin D insufficiency; reduced vitamin D activity (exacerbated by kidney dysfunction in nephropathy) can impair insulin release and increase inflammation (Orrù et al., 2020). In diabetic nephropathy, the kidneys become damaged, losing their ability to activate vitamin D, which leads to a calcium-phosphate imbalance and secondary hyperparathyroidism. Clinically, vitamin D deficiency is more prevalent in individuals with diabetes and nephropathy. Low vitamin D levels correlate with higher albuminuria and inflammatory markers, indicating a greater risk of nephropathy (Obaid et al., 2024). Notably, vitamin D has immunomodulatory effects that attenuate pro-inflammatory cytokines (e.g., it can suppress IL-6 and IL-12 production) (Ghaseminejad-Raeini et al., 2023). Some evidence suggests a complex relationship between vitamin D and BDNF. One study in people with type 2 diabetes found that low vitamin D status was associated with elevated BDNF levels, and subsequent vitamin D supplementation lowered BDNF levels while improving glycemic control. This inverse correlation suggests that vitamin D may influence neurotrophic pathways, although the underlying mechanisms remain under investigation (Alqudah et al., 2022).

Vitamin B12 and folate are crucial for neuronal health and one-carbon metabolism. Deficiencies in either vitamin, common in people with diabetes on metformin or with poor diets, can lead to elevated homocysteine levels (Mokgalaboni et al., 2024). Indeed, patients with type 2 diabetes are prone to hyperhomocysteinemia primarily due to suboptimal B12 and folate status. Elevated homocysteine is toxic to vessels and has pro-inflammatory effects, contributing to endothelial dysfunction and cardiovascular risk (Mokgalaboni et al., 2024). In diabetic nephropathy, reduced renal clearance further raises homocysteine, compounding this risk. Folate supplementation can significantly lower homocysteine in diabetic patients, potentially mitigating vascular injury (Mokgalaboni et al., 2024). Both B12 and folate are also essential for maintaining the nervous system; deficiencies may cause neuropathy, which can worsen diabetes-related nerve damage. While direct links to BDNF are not fully elucidated, adequate B12 and folate likely support neuronal repair pathways (Mathew et al., 2024). By preventing homocysteine accumulation and neurotoxicity, these vitamins may help preserve the microvascular and neural environment in which BDNF operates.

Zinc is an essential trace element involved in insulin synthesis, storage, and secretion, and it serves as a cofactor for antioxidant enzymes. Diabetic patients often exhibit zinc deficiency, caused by excessive urinary zinc losses and reduced gut absorption (Nie et al., 2022). In a study of diabetic nephropathy patients, over half had low serum zinc, and the low-zinc group showed more severe kidney disease pathology. Zinc deficiency in individuals with diabetes is associated with poorer glycemic control, increased oxidative stress, and impaired immune function. Conversely, sufficient zinc protects pancreatic β-cells from oxidative damage and induces antioxidant defenses through Nrf2 activation, potentially slowing the progression of nephropathy (Nie et al., 2022). Zinc also plays a neurotrophic role. Supplementation with zinc has been shown to increase circulating BDNF levels and improve mood in clinical trials, reflecting zinc’s support of neuronal health. Thus, zinc insufficiency in people with diabetes might not only exacerbate inflammation and insulin dysfunction but could also diminish BDNF-mediated neural protection. Ensuring adequate zinc may benefit both the metabolic and nervous system components of diabetes (Solati et al., 2015).

These factors form a tightly connected network that shapes diabetic outcomes: vitamin D status influences both BDNF and inflammatory cytokines. When vitamin D is sufficient, it suppresses the production of IL-6 and IL-12 and may boost BDNF. Conversely, when deficient, it fuels inflammation that downregulates BDNF (Alqudah et al., 2022; Fenercioglu, 2024; Zhang et al., 2017; Hsu, Sheu & Lee, 2023). Vitamins B1 2 and folate lower homocysteine and oxidative stress, thereby fostering a metabolic environment in which BDNF can exert its neuroprotective effects (Maryoud et al., 2023). Zinc similarly curbs oxidative damage and inflammation in diabetes, and supplementation raises BDNF levels, supporting neuronal health (Alqudah et al., 2022; Solati et al., 2015). By contrast, chronically elevated IL-6 and IL-12 in poorly controlled diabetes inhibit BDNF expression, as ongoing inflammation impairs growth factor signaling (Hsu, Sheu & Lee, 2023). Importantly, BDNF feeds back on the immune system, as higher BDNF levels promote the production of anti-inflammatory cytokines and further suppress IL-6 and IL-12 in stressed tissues. This reciprocal loop means that improving one element—through nutritional or anti-inflammatory interventions—can increase BDNF levels, which then further dampen inflammation (Alqudah et al., 2022). In summary, diabetic patients (with or without nephropathy) face a complex interplay of micronutrient deficiencies, inflammatory stress, and neurotrophic imbalance. Vitamins D, B1 2, folate, zinc, and calcium each protect metabolism and curb inflammation; IL-6 and IL-12 drive pathogenic inflammation; and BDNF mediates neural (and possibly renal) health. Their interactions—vitamin D curbing IL-6, zinc boosting BDNF, BDNF restraining cytokines, and so on—highlight a network of therapeutic targets. Understanding these links could enable the development of integrated strategies to manage diabetes and prevent complications, such as nephropathy, by simultaneously addressing micronutrient status, inflammation, and neurotrophic support (Alqudah et al., 2022; Hsu, Sheu & Lee, 2023).

The current study aimed to evaluate BDNF levels in diabetic patients with nephropathy and those without complications compared to healthy control subjects, and also to investigate the correlation between BDNF levels and other biochemical parameters, including vitamin elements and inflammatory markers, in Iraqi diabetic patients.

Materials & Methods

Study design and settings

An observational, case-control study included healthy control participants (n = 62) and patients with type 2 diabetes (n = 124). Type 2 diabetic patients are divided into two groups: diabetic patients with nephropathy (DN; n = 62) and diabetic patients without any complications (DM, n = 62). The diagnosis of diabetes was established according to the American Diabetes Association’s “Standards of Care in Diabetes” (American Diabetes Association Professional Practice Committee, 2023a) by a physician specializing in diabetes. The control group did not suffer from any disease. All patients and control subjects fasted overnight for about 12 h before taking the blood sample.

The study was conducted at the outpatient clinic of the National Diabetes Center, Mustansiriyah University, Baghdad, Iraq, and commenced on October 1, 2024, with completion on January 30, 2025.

In addition to the case groups, 62 healthy control volunteers were enrolled from among hospital staff and community volunteers at Mustansiriyah University. Before inclusion, each control completed a standardized medical-history questionnaire and underwent a focused physical examination by a diabetologist. Laboratory screening included fasting plasma glucose, HbA1c, serum creatinine (to estimate eGFR), lipid profile, and urinalysis. Individuals were excluded if they had any history or laboratory evidence of diabetes, hypertension, chronic kidney disease, cardiovascular disease, autoimmune disorders, or were taking any chronic medications. Only volunteers with normal physical findings and normal screening laboratories were retained as controls, ensuring the absence of chronic disease.

According to “Standards of Care in Diabetes” in the American Diabetes Association guidelines in 2023, levels of glycated hemoglobin A1c ≥6.5 are classified as diabetic mellitus (new onset) (ElSayed et al., 2023), while nephropathy, defined as using spot urinary albumin-to-creatinine ratio with albuminuria levels ≥300 mg/g creatinine (American Diabetes Association Professional Practice Committee, 2023b).

The current study’s inclusion criteria included adults newly diagnosed with type 2 diabetes in Iraq, regardless of sex. In the control group, volunteers were required to have fasting plasma glucose < 100 mg/dL and HbA1c < 5.7%, in addition to normal history, exam, and routine labs. An oral glucose tolerance test was not performed. The exclusion criteria were pregnant women, children, patients’ refusal to participate, patients on chronic medications (including antidiabetic, antihypertensive drugs), patients on chronic immunosuppressive drugs, patients with cancer and autoimmune disease, and patients with heart failure and other heart diseases.

Ethical approval

The Scientific and Ethical Committee approved this study at the College of Pharmacy, Mustansiriyah University (Approval number: 76, Research number: 89, Approval date: August 20, 2024). Written informed consent was obtained from all participants, and the study was conducted in accordance with the Declaration of Helsinki (1964) and its subsequent amendments. The study was reported according to STROBE guidelines for reporting observational studies (Von Elm et al., 2007).

Data collection

Information was collected from the patients through a structured questionnaire, which included their age. Their height and weight were measured directly by trained staff using a calibrated stadiometer and digital scale to minimize reporting errors in body-mass index (BMI) calculations. Blood pressure was also measured for all patients and healthy control subjects.

Sample collections

Fifteen milliliters of venous blood were collected from each participant by standard venipuncture into appropriate collection tubes. The blood was placed in a gel tube, left to clot for 15 min, and then centrifuged at 3,000 rpm for 15 min. Then, the serum samples were stored in Eppendorf tubes (Eppendorf AG, Hamburg, Germany) for freezing at −20 °C for further analysis.

Measurements of serum biochemical parameters and immunological markers

Serum BDNF levels (Catalog Number: ELH-BDNF, Human BDNF ELISA Kit, RayBiotech, Peachtree Corners, GA, USA) were measured using a sandwich ELISA based on the colorimetric method, as specified by the manufacturer.

Serum IL-12 (Catalog Number: ELH-IL12P70, Human IL-12 p70 ELISA Kit, RayBiotech, Peachtree Corners, GA, USA) and IL-16 (Catalog Number: ELH-IL16, Human IL-16 ELISA Kit, Peachtree Corners, GA, USA) levels were measured using sandwich ELISA based on the colorimetric method, according to manufacturer specifications (Waheed, 2017; Waheed et al., 2020).

The total cholesterol (reference number: TK41021, SPINREACT, Sant Esteve de Bas, Girona, Spain), triglyceride (reference number: MX41031, SPINREACT, Sant Esteve de Bas, Girona, Spain), HDL-cholesterol (reference number: 1001095, SPINREACT, Sant Esteve de Bas, Girona, Spain), urea (reference number: TK41041, SPINREACT, Sant Esteve de Bas, Girona, Spain), serum creatinine (reference number: MD1001111, SPINREACT, Sant Esteve de Bas, Girona, Spain), serum calcium (reference number: MD1001065, SPINREACT, Sant Esteve de Bas, Girona, Spain), and serum zinc (reference number: 1001350, SPINREACT, Sant Esteve de Bas, Girona, Spain), according to manufacturer specifications (Al-Ezzi, Mohsin & Waheed, 2020).

Serum folic acid and serum vitamin B12 were measured using the VIDAS technique (BioMérieux, Marcy-l’Étoile, France). Hemoglobin A1c was measured by using spectrophotometers (Human Co., Wiesbaden, Germany) (Majid, Abbas & Waheed , 2022). Vitamin D3 levels were measured using CL-900i (Shenzhen Mindray Bio-Medical Electronic, Shenzhen, China).

Sample size calculations

G-power was used to calculate the sample size, the F test family was used with an effect size of 0.25, α = 0.05, 85% power (β = 0.15), and we arrived at 184 patients divided into three groups (each includes 62 patients).

Statistical analysis

The Anderson-Darling test assessed whether continuous variables follow a normal distribution. One-way ANOVA is used to analyze the differences between more than two groups (if they follow a normal distribution with no significant outlier); after that, if the results are significant, the post hoc Tukey test will be used to find which pair is significant; if the variables do not follow a normal distribution, Kruskal Wallis test was performed with post hoc Dunn test for pairwise comparison. Linear regression analysis was performed to assess the relationship between different variables. Spearman’s correlation was used because most of the data did not follow a normal distribution. The receiver operating characteristic (ROC) curve is used to assess the validity of different parameters in distinguishing between active cases and control (negative) cases. The area under the curve, i.e., AUC and its p-value, prescribes this validity (if AUC ≥ 0.9 means excellent test, 0.8–0.89 means good test, 0.7–0.79 fair test, otherwise unacceptable). The trapezoidal method was used to calculate the curve.

Minitab 17.1.0, MedCalc Statistical Software version 14.8.1 (MedCalc Software bvba, Ostend, Belgium; 2014), GraphPad Prism version 10.4.1 for Windows, GraphPad Software, San Diego, California, USA, a software package used to make the statistical analysis, p-value considered when appropriate to be significant if less than 0.05.

Results

The characteristics of the patients and subjects are shown in Table 1. Both patient groups matched in age and BMI with the healthy group, and there was no significant difference between them [p > 0.05]. It was noted that there was a significant difference between the groups of patients and healthy individuals in terms of blood pressure for SBP and DBP [p < 0.05].

Table 1 Assessment of demographic characteristics.

Variables	Control	DM	DN	p-value	
Number	62	62	60	–	
Age (years), mean ± SD	57.3 ± 4.4	58.0 ± 6.0	58.8 ± 4.1	0.242	
BMI (kg/m2), mean ± SD	25.29 ± 2.12	25.79 ± 1.99	25.43 ± 2.07	0.387	
Sex				0.851	
Female	32 (51.6%)	34 (54.8%)	34 (56.7%)	
Male	30 (48.4%)	28 (45.2%)	26 (43.3%)	
SBP (mmHg), mean ± SD	123 ± 6	127 ± 14	163 ± 19	<0.001	
DBP (mmHg), mean ± SD	69 ± 7	82 ± 9	102 ± 2	<0.001	
Notes.

DM diabetes mellitus

DN diabetic nephropathy

BMI body mass index

SBP systolic blood pressure

DBP diastolic blood pressure

SD standard deviation

Serum BDNF was significantly lower in the DN group (27.23 ±  7.81 ng/ml) when compared to the DM group (34.1 ±  7.28 ng/ml) [p < 0.001], also when compared with the control group (42.10 ± 6.42 ng/ml) [p < 0.001]. Both FPG and HbA1c were significantly higher in the DN group (median, 117.5 & 8.5 mg/dL, respectively) when compared to both DM (median, 115 & 7.6 mg/dL, respectively) and control groups (median, 117.5 & 8.5 mg/dL, respectively), as seen in Table 2.

Table 2 Assessment of various biomarker levels.

Variables	Control	DM	DN	p-value	
Number	62	62	60	–	
BDNF (ng/ml)a	42.10 ± 6.42	34.1 ± 7.28	27.23 ± 7.81	<0.001	
FPG (mg/dL)b	84 (81–87)	115 (104–125)	117.5 (105–132)	<0.001	
HbA1c (%)b	5.5 (5.3–5.7)	7.6 (7.4–8.2)	8.55 (8.2–9.5)	<0.001	
Cholesterol (mg/dL)b	184 (170–192)	199 (178–239)	214.5 (196–307)	<0.001	
TG (mg/dL)b	172 (164–182)	179 (162–193)	186.5 (172–231)	<0.001	
HDL (mg/dL)b	43 (40–48)	41 (38–43)	43 (38–46)	0.010	
Blood urea (mg/dL)b	33 (31–37)	32 (28–37)	54.5 (49–61)	<0.001	
Serum creatinine (mg/dL)b	0.8 (0.7–0.9)	0.8 (0.7–0.9)	1.8 (1.7–2.1)	<0.001	
Serum folic acid (ng/mL)b	14 (11–16)	14 (11–17)	9 (5–11)	<0.001	
Vitamin B12 (μ g/L)b	37 (34–44)	37 (34–41)	27 (22–32)	<0.001	
Total serum calcium (mg/dL)b	8.5 (7.8–9.2)	8.6 (7.7–9.3)	8.6 (7.6–9.3)	0.880	
Serum Zn (µmol/L)b	92 (86–96)	89 (80–93)	83.5 (78–91)	<0.001	
Vitamin D3 (ng/mL)b	25 (18–27)	21 (14–25)	16 (12–21)	<0.001	
IL-6 (pg/mL)b	8.4 (6.9–10.1)	12.6 (10.6–14.6)	15.3 (13.2–17.5)	<0.001	
IL-12 (pg/mL)b	7.4 (6.4–8.3)	8.5 (7.5–11.4)	11.3 (7.4–14.3)	<0.001	
Notes.

a Data presented as mean ± standard deviation.

b Data presented as median (interquartile range).

DM diabetes mellitus

DN diabetic nephropathy

TG triglyceride

HDL high-density lipoprotein

FPG fasting plasma glucose

BDNF brain-derived neurotrophic peptide

IL interleukin

There is a significant difference in folic acid levels in the DN group (median, nine ng/mL) when compared to both DM (14 ng/mL) and control (median, 14 ng/mL) [p < 0.001]. Still, no significant difference exists between the control group and DM [p > 0.05], as seen in Fig. 1D. There is a significant difference in vitamin B12 levels in the DN group (median, 27 ng/L) when compared to both DM (median, 37 ng/L) and control (median, 37 ng/L) [p < 0.001]. Still, no significant difference exists between the control group and DM [p > 0.05], as seen in Fig. 1E. There was a significant difference in serum zinc levels when comparing the DN group (median, 83.5 µmol/L) with the control group (median, 92 µmol/L). There was no significant difference between nephropathy and diabetic patients (median, 89 µmol/L), as seen in Fig. 1F. Serum vitamin D was significantly lower in the DN group (median, 16 ng/ml) when compared to the DM group (median, 21 ng/ml) [p < 0.001] and also when compared with control (median, 25 ng/ml) [p <  0.001], as seen in Fig. 1G.

Figure 1 Graphical presentations of serum levels of various markers.

(A) BDNF, (B) HbA1c, (C) FPG, (D) folic acid, (E) vitamin B12, (F) zinc, (G) vitamin D, (H) IL-6, and (I) IL-12.

Regarding the inflammatory markers, IL-6 and IL-12 significantly higher in the DN group (median, 15.3 & 11.3 pg/mL, respectively) compared to the control group (median, 8.4 & 7.4 pg/mL, respectively); at the same time, IL-6 was significantly higher in DN group compared to the DM group, as seen in Figs. 1H and 1I.

For the control group, serum BDNF is directly correlated with zinc (Zn) and IL-12 (r = 0.35, r = 0.29, respectively). However, as seen in Fig. 2, BDNF is inversely correlated with age (r = −0.40).

Figure 2 Correlation matrix heatmap of various biomarkers in the control group (blue color indicates direct correlation, while red color indicates inverse correlation; value represents correlation coefficient, and value above 0.25 indicates significant correlation).

In the DM group, there is a direct correlation between BDNF and HbA1c and urea (r = 0.26, r = 0.35, respectively). There is also an inverse correlation between BDNF and FPG (r = −0.30), as shown in Fig. 3.

Figure 3 Correlation matrix heatmap of various biomarkers in DM group (blue color indicates direct correlation, while red color indicates inverse correlation; value represents correlation coefficient, and value above 0.25 indicates significant correlation).

For the DN group, a direct correlation exists between BDNF and FPG (r = 0.31) and serum creatinine (r = 0.27), as shown in Fig. 4.

Figure 4 Correlation matrix heatmap of various biomarkers in DM with nephropathy group.

The Area Under the Curve (AUC) was calculated to determine which studied biomarkers are more sensitive and specific. The AUC value for BDNF in the DN group vs. control group was 0.938, indicating excellent ability to discriminate, with high sensitivity (93.33%) and specificity (83.87%); when comparing between DN vs. DM group, the AUC for BDNF was 0.738, indicating fair ability to discriminate, with sensitivity (80%) and specificity (58.06%) (Table 3 & Fig. 5).

Table 3 Diagnostic utility analysis of the discriminate diabetic nephropathy from control or DM alone.

Groups	AUC (95%CI)	p-value	Cut-off	SN	SP	PPV	NPV	
DN vs. control	0.938 (0.879 to 0.973)	<0.001	≤36	93.33	83.87	84.8	92.9	
DN vs. DM	0.738 (0.650 to 0.813)	<0.001	≤32	80.00	58.06	64.9	75.0	
Notes.

DM diabetes mellitus

DN diabetic nephropathy

SN sensitivity

SP specificity

PPV positive predictive value

NPV negative predictive value

AUC area under the curve

CI confidence interval

Figure 5 ROC curves showing the ability of BDNP to diabetic nephropathy from (A) control or (B) DM alone.

Discussion

Nephropathy impairment is one of the most common diseases that accompany diabetes. In the current study, serum BDNF significantly decreased in diabetic nephropathy patients compared to diabetic patients without complications and the healthy control group. Diabetes patients face an increased risk of neurological issues; despite the many advances in elucidating pathophysiology and developing therapies, diabetes remains a challenging disease to manage (Sugandh et al., 2023). In recent years, it has become clear that diabetes may be associated with a higher incidence of neurological issues because of the impact of blood glucose levels on neurogenesis and neuroprotection through the modulation of brain-derived neurotrophic factors (Chavda et al., 2024).

Although several reports have examined serum BDNF in diabetes, few have systematically explored its relationships with vitamins D, B12, and folic acid, trace elements (zinc, calcium), and pro-inflammatory cytokines (IL-6, IL-12) in the context of diabetic nephropathy. By concurrently profiling this comprehensive panel in type 2 diabetic patients with and without nephropathy, our work fills a critical gap, providing the first integrated assessment of neurotrophic, micronutrient, and immunological interactions in diabetic kidney disease.

The potential relationship between diabetes and serum BDNF suggests a possible interaction between diabetes and brain function (He et al., 2024). It is well established that general biological processes, such as hyperglycemia and kidney function disorders in type 2 diabetes mellitus, can lead to alterations in the concentration of BDNF in serum (Dong et al., 2024). Diabetic patients exhibited significantly lower serum BDNF levels compared to controls, consistent with prior studies. However, our cross-sectional design only demonstrates correlation, not causation; whether strategies to elevate BDNF can prevent diabetic complications remains to be tested in prospective or interventional trials.

Oxidative stress and inflammation play an important pathogenic role in the interaction of diabetes and depression, which provides a feasible explanation of the role of diabetes in changes in serum concentrations of BDNF (Correia, Cardoso & Vale, 2023). It is particularly important to pay attention to maintaining optimal levels of BDNF to protect neurons from damage in patients with diabetic nephropathy (Tanase et al., 2025).

Low levels of BDNF in patients with diabetic nephropathy may be related to several mechanisms, including, as mentioned earlier, inflammation and oxidative stress, because some interleukins inhibit the production of this protein (such as IL-6 and IL-12) (Dawi et al., 2025). It may also result from another mechanism, insulin resistance, since insulin signals play an important role in regulating the production of BDNF (Harvey & Rios, 2024). Among the proposed mechanisms is the result of a defect in the lining of blood vessels and a decrease in the transfer of BDNF, since diabetic nephropathy is accompanied by damage to the microvasculature, which weakens the transfer of BDNF (Chiang et al., 2024). One of the important reasons we mentioned earlier is that it is due to a decrease in kidney function, which leads to an increase in the loss of large proteins, including BDNF (Gliwińska et al., 2024).

Based on the current study, a relationship between BDNF is documented through the presence of a direct association with urea and HbA1c in a group of diabetic patients, as well as a direct relationship between the levels of BDNF in patients and serum creatinine. Meta-analyses show that diabetic patients featured lower BDNF levels than control participants (Moosaie et al., 2023). The results of a small study also indicated that the nadir BDNF level predicted a more significant decline in eGFR (Hsu, Sheu & Lee, 2023). In addition, they found that type 2 diabetic patients exhibited a significantly lower BDNF level compared with the control group. There was a significant negative correlation between serum BDNF levels and a larger amount of albuminuria output. Patients with a larger amount of albuminuria are more likely to have significantly lower BDNF levels. Among the enrolled patients, those experiencing macroalbuminuria showed lower BDNF levels than the other groups (Hsu, Sheu & Lee, 2023).

The interconnections between micronutrient status, inflammatory activity, and BDNF are crucial for understanding the progression of diabetes. Poor nutritional status in diabetes, such as deficiencies in vitamin D, vitamin B12, folate, or zinc, can exacerbate inflammation. For instance, vitamin D deficiency removes the anti-inflammatory brake, leading to higher levels of IL-6 and TNF-α and accelerating renal injury (He et al., 2025). Low zinc similarly permits greater inflammatory cytokine release via NLRP3 inflammasome activation. Folate or vitamin B12 deficiency elevates homocysteine, which directly damages the endothelium and triggers oxidative stress (Zhu et al., 2025). Thus, deficiencies create a pro-inflammatory milieu that worsens diabetic complications.

In the current study, serum levels of IL-6 and IL-12 were significantly higher in the DN compared to the DM and control groups. At the same time, BDNF levels were significantly lower in the DN group compared to the DM and control groups, indicating an inverse relationship between inflammatory markers and BDNF. In diabetes, chronic inflammation—with elevated IL-6, IL-12, IL-1β, and other cytokines—directly suppresses BDNF expression and signaling. Research on chronic kidney disease consistently shows that higher pro-inflammatory cytokine levels are associated with lower BDNF levels (Ferreira et al., 2024; Hsu, Sheu & Lee, 2023). Persistently raised IL-6, for example, is known to block neurotrophic pathways and neurogenesis (Kummer et al., 2021; Erta, Quintana & Hidalgo, 2012). As a result, conditions like diabetes and atherosclerosis typically present both heightened inflammation and reduced BDNF (Lima Giacobbo et al., 2019). In diabetic nephropathy, this ongoing inflammatory milieu likely blunts BDNF production in the brain and throughout the body. That sets up a vicious cycle: inflammation lowers BDNF, diminished BDNF aggravates endothelial and neuronal injury, and the resulting tissue damage fuels even more inflammation.

In the current study, serum zinc levels were lower in DN compared to the other groups. In the present study, Zn showed a direct correlation with BDNF (r = 0.35). Adequate levels of certain micronutrients are needed for normal BDNF-mediated neuroprotective effects. Zinc in particular is required for proper BDNF signaling in the brain; experimental zinc deficiency leads to impaired BDNF signaling and cognitive dysfunction (Yang et al., 2013). In diabetes, ensuring zinc sufficiency might help preserve BDNF activity and mitigate neuropathic changes. In the present study, vitamin D3 levels were lower in DN, followed by DM and the control groups, indicating the direct relationship between vitamin D and BDNF. Vitamin D has complex interactions with BDNF; acute changes in vitamin D status can alter BDNF levels (as observed in one study) (Alqudah et al., 2022). Overall, vitamin D’s neuroprotective, anti-inflammatory roles likely support a healthier environment for BDNF function. Vitamin B12 and folate contribute to neuronal health by facilitating myelin synthesis and reducing homocysteine, indirectly supporting BDNF’s neurotrophic actions by maintaining neuronal integrity (Sharma & Aran, 2025; Umekar et al., 2025). In summary, better nutritional status tends to correlate with higher BDNF and lower inflammation, whereas deficiencies correlate with a loss of neurotrophic support and heightened inflammation.

BDNF itself can influence metabolic-inflammation interplay. Higher BDNF levels improve insulin sensitivity and reduce blood glucose, which can then lower glucotoxic stress and inflammatory drive in tissues (Tonra et al., 1999). BDNF also modulates immune cells; some studies suggest BDNF can dampen inflammatory activation in certain contexts (Xiong et al., 2024; Parrott et al., 2021). Conversely, when BDNF is low, metabolic control worsens (higher glucose, more AGE formation), leading to more oxidative stress and cytokine release (Rodziewicz et al., 2020; Delezie et al., 2019; Tanase et al., 2025). Thus, BDNF is a key intermediary: it is affected by nutrition and inflammation, and in turn affects metabolic and inflammatory outcomes. The study’s observed correlations likely reflect this web of interactions. For example, patients with higher vitamin D or zinc levels (indicating better nutrition) may have had higher BDNF and lower IL-6 levels. In contrast, those with intense inflammation (characterized by high IL-6/IL-12 ratios) showed diminished BDNF levels. This interplay highlights that addressing one aspect (such as supplementing vitamins or reducing inflammation) can have a beneficial impact on others.

Generally, BDNF is affirmed to play an important role in nerve protection in various ways. The renal nerve layer is sensitive to the protective effects of BDNF. Nevertheless, this protective mechanism is weakened by increasing albuminuria output and macroalbuminuria (Sun et al., 2018). Consequently, a growing number of tests confirm the existence of a relationship between the decline in BDNF and the development of DN (Pisani et al., 2023). These studies support the notion that a gradually decreasing BDNF level may exacerbate the severity of kidney function due to the progression of DN (Hsu, Sheu & Lee, 2023). With the development of the most recent research findings, DN is accompanied by declining serum BDNF levels. Such a decrease in the BDNF level is considered to play an important role in the severity of renal impairment in the diabetic population. With increasing albuminuria output, the protective effect of BDNF on the renal nerve is gradually removed (Gliwińska et al., 2024). In the event of macroalbuminuria, it is observed that serum BDNF levels are significantly reduced (Burrows, Koyama & Pavkov, 2022).

Study strengths

Our study provides a comprehensive biochemical and immunological profile, simultaneously assessing BDNF, vitamins D, B12, folic acid, zinc, calcium, IL-6, and IL-12, in well-matched groups of type 2 diabetic patients with and without nephropathy, as well as healthy controls. Employing validated ELISA and automated assays increases measurement reliability, while strict inclusion criteria and standardized sample handling minimize pre-analytical variability. Additionally, the study’s strengths include a broad panel of neuro-metabolic and inflammatory markers in a single cohort, rigorous case–control matching by age, sex, and BMI, and the use of standardized, commercially available assay kits.

Study limitations

Cross-sectional, single-time-point design precludes causal inference and fails to capture intra-individual variability in micronutrients or proteinuria. The control group, drawn from university and hospital staff, may not fully represent the general population. Lack of data on dietary intake, sun exposure, or seasonal variation in vitamin D and zinc status. A moderate sample size and single-center setting limit the statistical power for subgroup analyses.

Because no oral glucose tolerance test was performed, early dysglycemia may have gone undetected in controls, potentially attenuating between-group contrasts. Likewise, reliance on a single measurement of proteinuria and renal markers cannot capture temporal fluctuations in nephropathy severity. Future longitudinal studies incorporating oral glucose tolerance test (OFTT) screening and serial renal assessments are warranted to reduce misclassification bias and better delineate the interplay between BDNF, micronutrients, inflammation, and kidney function in diabetes.

Conclusions

Our findings reveal that serum BDNF levels are significantly reduced in type 2 diabetic patients with nephropathy compared to both uncomplicated diabetics and healthy controls. These declines correlate with deficiencies in vitamins D, B12, and folate, as well as lower zinc levels and elevated IL-6/IL-12 levels. These data suggest that BDNF may serve as a cross-sectional biomarker of neuro-metabolic derangement in diabetic kidney disease. However, prospective studies are required to define a ‘normal’ reference range for BDNF, to investigate whether early BDNF monitoring can predict the onset of diabetic neuropathy, and to determine whether interventions that increase BDNF reduce the risk of complications.

The study highlights significant differences in vitamin D, vitamin B12, folate, zinc, calcium, IL-6, IL-12, and BDNF between diabetic patients with and without nephropathy. Diabetic nephropathy is characterized by worsened nutritional deficiencies and heightened inflammation, which appear to go together with reduced BDNF-mediated support. This complex interplay suggests that treating diabetes requires more than just glycemic control, attention to nutritional status and inflammation, as well as support for neurotrophic factors, is essential to improve outcomes and prevent the progression of nephropathy and other complications. The findings reinforce the concept that better nutritional and inflammatory management may preserve BDNF and protect patients, a holistic approach that could slow the vicious cycle of diabetes and its complications.

Supplemental Information

Supplemental Information 1 Data set of BDNF

Supplemental Information 2 STROBE checklist

We want to thank Mustansiriyah University and its affiliated institutions (College of Pharmacy and the National Diabetes Center) for their contributions to completing the research, including obtaining volunteer patients and conducting laboratory tests.

Additional Information and Declarations

Competing Interests

Author Contributions

Human Ethics

Data Availability

The authors declare there are no competing interests.

Huda Jaber Waheed conceived and designed the experiments, performed the experiments, analyzed the data, prepared figures and/or tables, authored or reviewed drafts of the article, and approved the final draft.

Nawfal A. Numan performed the experiments, authored or reviewed drafts of the article, and approved the final draft.

The following information was supplied relating to ethical approvals (i.e., approving body and any reference numbers):

The Scientific and Ethical Committee approved this study at the College of Pharmacy, Mustansiriyah University (Approval number: 76, Research number: 89).

The following information was supplied regarding data availability:

The data is available in the Supplementary Files and at Zenodo: Waheed, H. J. (2025). BDNP patients data [Data set]. Zenodo. https://doi.org/10.5281/zenodo.14910870.

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
