# Peer review of "Serum brain-derived neurotrophic factor, micronutrient status, and inflammatory cytokines in type 2 diabetes with nephropathy: a case–control analysis"

_PeerJ, doi:10.7717/peerj.20086_

## Round 0.1 · original submission · Major Revisions

**Language Note:** The review process has identified that the English language must be improved. PeerJ can provide language editing services - please contact us at [email protected] for pricing (be sure to provide your manuscript number and title). Alternatively, you should make your own arrangements to improve the language quality and provide details in your response letter. – PeerJ Staff

Reviewer 1 ·

Basic reporting

The subject matter discussed in the article is essential and current, and the tremendous effort put into the research should certainly be appreciated. The work is well structured, raw research data for verification are included, and the results are clearly presented in figures and tables.

However, significant changes should be made to the manuscript.

The article is written in English. Nevertheless, the English language should be improved to ensure that an international audience can clearly understand the text. For example, in the Results section and the Discussion section, the terms "increased" and "decreased" are used to describe the obtained results. However, since it is not a longitudinal assessment or an assessment following some intervention, the static term "was lower/higher" should be used instead. Having a colleague who is proficient in English and familiar with the subject matter review the manuscript, or contacting a professional editing service, would be reasonable.

The authors stated that the aim of the study was "to evaluate the BDNF levels in diabetic patients with nephropathy and diabetic patients without complication compared to healthy control subjects, and also to find the correlation between BDNF levels and other biochemical parameters, including vitamin elements and inflammatory markers, in Iraqi diabetic patients." However, the introduction lacks justification for this study's aim in the context of vitamins, zinc, and calcium. Moreover, this is not reflected in the study's title or, more importantly, discussed in the discussion part.

Experimental design

The paper presents the results of original primary research and is within the Aims and Scope of the journal. However methods are presented without sufficient detail and information to replicate.
1. It would be worthwhile to explain where the "healthy controls" were recruited from. Was the lack of chronic diseases in these people verified in any way?

2. It follows from the content of Section 2.1 that the diagnosis of diabetes was made in accordance with the ADA recommendations. At the same time, paragraph 2.3 states that only cases of new-onset diabetes, defined as an HbA1c level of 6.5% or higher, were considered. In such a case, we have no guarantee that the "healthy" group does not include people with carbohydrate metabolism disorders (e.g., there are cases with HbA1C=5.7, 5.8, 6.1%, in which case it is recommended to verify the diabetes status based on an oral glucose tolerance test) - this should be indicated as a weakness of the work.

3. Was the diagnosis of diabetic nephropathy based on only one measurement of albuminuria? Was a urinary tract infection excluded? These may also be weaknesses of the work.

4. It was stated that information on body weight and height was collected during an interview with the examined person - this information may be burdened with a fairly significant error in this situation.

5. The patient assessment lacks basic information about kidney function - i.e., glomerular filtration rate/GFR.

6. It is a pity that in the case of people with diabetes, the occurrence of other chronic diseases that may affect the BDNF level was not verified

7. In paragraph 2.6, it is written that "Fifteen ml of venous blood was drawn from the patients and the control group using a 10 ‎ml syringe." - These two pieces of information  do not correspond to each other, and they sound strange; maybe it is worth not giving the volume of the syringe - it is not essential,

Validity of the findings

1. In Table 1, please verify the correctness of the units for SBP and DBP. If they are given in mmHg, then the decimal point was misplaced.

2. Some results, although statistically significant findings, are not discussed at all in the discussion section. There is no reference in the discussion to the assessed relationship of BDNF with vitamins, zinc, and calcium. At the same time, the work lacks consideration of other factors and diseases that may be associated with BDNF levels (e.g., cognitive impairment, physical activity/sarcopenia, etc.)

3. There is no indication of the weaknesses of the work and strengths in the summary of the discussion, and reference to whether these research results can be generalized.

4. The statement summarizing the abstract and constituting the conclusion, "So, we can ‎conclude that early measurement of BDNF levels could identify diabetes patients ‎with normal ‎BDNF levels, thus protecting them from developing diabetic neuropathy," is not justified and cannot be based on the results of the study presented in the work. It is not clear what "normal BDNF levels" mean in light of the studies conducted, and how the finding of these "normal BDNF levels" protects against the development of diabetic neuropathy?

5. Similarly, the sentence in the Discussion section: "The results clearly show that patients with diabetes ‎had lower levels of BDNF, which suggests that maintaining high BDNF levels while being idle may ‎reduce the incidence of complications in diabetic patients. " is an unjustified statement. The study assessed correlations; it is not known what the cause-and-effect relationships are in this area.

6. It would be worthwhile to indicate where the authors see further directions for research in this area.

The underlying data have been provided,

Additional comments

The researchers indeed conducted valuable research, but the presentation of the results from these studies needs refinement in several aspects. Firstly, the aims of the work, research hypotheses, and methodology should be more clearly and precisely described. Additionally, the discussion of the obtained results and the conclusions drawn should be more effectively presented, taking into account study limitations.

Reviewer 2 ·

Basic reporting

Ensure the manuscript is written in clear, fluent English. Strengthen the background section to provide a well-structured and informative context for the study. Confirm that all referenced literature is current, relevant, and appropriately cited to support the research objectives.

Chapter “2 Materials & Methods” should be reformatted. Subchapters “2.3 Definitions” and “2.4 Inclusion and exclusion criteria” should be introduced into subchapter “2.1 Study design and settings”. Additionally, the description of all instruments and kits used must include the country of manufacture for these instruments (e.g., Humen Co., line 129).

Figures are relevant, well labelled, and described. Raw data supplied according to PeerJ policy.

Experimental design

The novelty of the study lies in the fact that there are few studies focused on the correlation of BDNF with different clinical biomarkers in patients with diabetes, but not in diabetic nephropathy.

Although the groups used a minimal number of patients, they observed a significant decrease in BDNF in patients with diabetic nephropathy and a direct correlation between BDNF, FPG, and serum creatinine.

Validity of the findings

Their conclusion related to BDNF levels decreasing to a degree lower than normal in diabetics and more so in diabetic complications such as nephropathy, thus being able to identify diabetic patients with an increased risk of developing diabetic neuropathy is supported by the results.

---

## Round 0.2 · accepted · Accept

All issues pointed out by the reviewers were addressed, and the manuscript was substantially revised. The amended version is acceptable now.